# Analytical and diagnostic performance characteristics of reverse-transcriptase loop-mediated isothermal amplification assays for dengue virus serotypes 1–4: A scoping review to inform potential use in portable molecular diagnostic devices

Paul Arkell[1]*, Dumrong Mairiang[2,3], Adisak Songjaeng[2,4], Kenny Malpartida-Cardenas[1], Kerri Hill-Cawthorne[1], Panisadee Avirutnan[2,3,4], Pantelis Georgiou[1,5], Alison Holmes[1,6], Jesus Rodriguez-Manzano[1]

1 Centre for Antimicrobial Optimisation, Department of Infectious Disease, Imperial College London, Hammersmith Hospital, London, United Kingdom, 2 Siriraj Center of Research Excellence in Dengue and Emerging Pathogens (SiCORE-Dengue), Faculty of Medicine Siriraj Hospital, Mahidol University, Bangkok, Thailand, 3 Molecular Biology of Dengue and Flaviviruses Research Team, Medical Molecular Biotechnology Research Group, National Center for Genetic Engineering and Biotechnology (BIOTEC), National Science and Technology Development Agency (NSTDA), Bangkok, Thailand, 4 Division of Dengue Hemorrhagic Fever Research, Department of Research and Development, Faculty of Medicine Siriraj Hospital, Mahidol University, Bangkok, Thailand, 5 Department of Electrical and Electronic Engineering, Imperial College London, London, United Kingdom, 6 David Price Evans Global Health and Infectious Disease Research Group, University of Liverpool, Liverpool, United Kingdom

* p.arkell@imperial.ac.uk

## Abstract

Dengue is a mosquito-borne disease caused by dengue virus (DENV) serotypes 1–4 which affects 100–400 million adults and children each year. Reverse-transcriptase (RT) quantitative polymerase chain reaction (qPCR) assays are the current gold-standard in diagnosis and serotyping of infections, but their use in low-middle income countries (LMICs) has been limited by laboratory infrastructure requirements. Loop-mediated isothermal amplification (LAMP) assays do not require thermocycling equipment and therefore could potentially be deployed outside laboratories and/or miniaturised. This scoping literature review aimed to describe the analytical and diagnostic performance characteristics of previously developed serotype-specific dengue RT-LAMP assays and evaluate potential for use in portable molecular diagnostic devices. A literature search in Medline was conducted. Studies were included if they were listed before 4th May 2022 (no prior time limit set) and described the development of any serotype-specific DENV RT-LAMP assay ('original assays') or described the further evaluation, adaption or implementation of these assays. Technical features, analytical and diagnostic performance characteristics were collected for each assay. Eight original assays were identified. These were heterogenous in design and reporting. Assays' lower limit of detection (LLOD) and linear range of quantification were comparable to RT-qPCR (with lowest reported values $2.2 \times 10^1$ and $1.98 \times 10^2$ copies/ml, respectively, for

**Data Availability Statement:** All data is included in supporting information files.

**Funding:** PA is funded by the Wellcome Trust (ref: 215688/Z/19/Z). KMP is funded by NIHR (ref: NIHR134694). KHC is funded by the Wellcome Trust (ref: 226691/Z/22/Z). PG and JRM are funded by Imperial College London. AH is funded by Imperial College London and the University of Liverpool. DM, AS and PA are funded by Mahidol University. The funders had no role in study design, data collection and analysis, decision to publish, or preparation of the manuscript.

**Competing interests:** The authors have declared that no competing interests exist.

studies which quantified target RNA copies) and analytical specificity was high. When evaluated, diagnostic performance was also high, though reference diagnostic criteria varied widely, prohibiting comparison between assays. Fourteen studies using previously described assays were identified, including those where reagents were lyophilised or 'printed' into microfluidic channels and where several novel detection methods were used. Serotype-specific DENV RT-LAMP assays are high-performing and have potential to be used in portable molecular diagnostic devices if they can be integrated with sample extraction and detection methods. Standardised reporting of assay validation and diagnostic accuracy studies would be beneficial.

## Background

Dengue is a mosquito-borne disease caused by dengue virus serotypes 1–4 (DENV 1–4). International travel, urbanisation and climate change have contributed to increasing global incidence with up to half the world's population across 125 tropical and sub-tropical countries now at-risk [1–3]. There are an estimated 100–400 million infections annually which cause a spectrum of disease ranging from asymptomatic or mild, self-limiting symptoms to severe forms of the disease, including dengue haemorrhagic fever and dengue shock syndrome [4, 5]. 'Secondary infection', when an individual is infected for a second (or subsequent) time in their life by a different serotype to their 'primary infection' is most likely to result in severe disease [6–8]. Therefore, shifts in the predominant circulating DENV serotype can be associated with outbreaks [9]. Better access to serotype-specific diagnostic testing for dengue may improve case-management, surveillance and disease control [10].

### The dengue diagnostic gap

Reverse-transcriptase polymerase chain reaction (RT-PCR) assays are the current gold-standard in diagnosis and serotyping of dengue infections [11]. These detect DENV ribonucleic acid (RNA) which is present in clinical samples from the onset of symptoms in both primary and secondary infection. A RT enzyme is used to synthesise complementary deoxyribonucleic acid (cDNA) from a target RNA sequence, and a DNA polymerase enzyme is used to amplify the cDNA. PCR primers can be designed to either bind regions of the DENV genome which are conserved across serotypes (resulting in a 'generic' dengue assay), or regions which are specific to an individual serotype (resulting in a serotype-specific assay) [12]. PCR assays can quantify DENV RNA if performed in real-time PCR (RT-qPCR), and can be multiplexed with assays for other pathogens [13, 14]. However, in all PCR assays, amplification of nucleic acid occurs during repeated cycles of heating and cooling to achieve precise denaturation and annealing temperatures, requiring significant laboratory infrastructure and a reliable power supply. They also require skilled operators and systems for quality and safety. As such, deploying PCR facility in many low-middle income countries (LMICs) is impractical. Patients in most settings do not receive serotype-specific DENV testing and this may impact disease surveillance and control efforts. A multicentre observational study on the global availability of testing highlighted the diagnostic gap in LMICs, which particularly affects remote and regional areas [15, 16].

### Loop-mediated isothermal amplification (LAMP) as a potential solution

Loop-mediated isothermal amplification (LAMP) is an isothermal nucleic acid amplification method first described in 2000 by Notomi et al. [17]. When coupled with a reverse-

transcriptase step (RT-LAMP), it can be used to detect RNA (Fig 2). Samples can be analysed directly, but usually undergo a nucleic acid extraction step to isolate and purify RNA, removing undesired components which may inhibit or otherwise affect the efficiency of the amplification reaction. LAMP uses a *Bacillus stearothermophilus* (*Bst*) DNA polymerase enzyme which possesses high autocycling strand displacement activity, allowing DNA synthesis to occur at a constant temperature (typically between 60–65 degrees Celsius). Multiple primers are used to create continuous loop structures during amplification. Primers may be designed to produce either generic or serotype-specific dengue assays. LAMP primers include forward and backward outer primers (F3 and B3, respectively), forward and backward inner primers (FIP and BIP), and forward and backward loop primers (FLP and BLP, which are not essential but can improve efficiency of the reaction). Amplified nucleic acids can be detected and/or quantified using various methods including visual inspection, turbidometry, gel electrophoresis, real-time monitoring using intercalating dyes and hybridisation with fluorescent probes, or non-fluorescence methods such electrochemical sensors [18–21]. Because LAMP assays do not require thermocycling equipment, they have long been considered potentially more suited to miniaturisation and/or deployment outside the laboratory setting than PCR assays, including for dengue and other neglected tropical diseases [22, 23].

## Evaluating RT-LAMP assay performance

Studies which evaluate RT-LAMP assays may include measurement of various analytical performance characteristics (i.e. those which are inherent to the assay) and/or diagnostic (clinical) performance characteristics (i.e. those which become relevant when the assay is used to detect a condition or disease) [24–26].

Analytical performance characteristics include lower limit of detection (LLOD, also known as 'analytical sensitivity' which is defined as the lowest concentration of a given substance that can be detected) and the linear range of quantification (also known as the 'reportable range', which for quantitative assays is the span of test result values over which the accuracy of the measurement can be verified). Additionally, analytical specificity (which is the ability of an assay to detect only the intended target and the absence of 'cross-reaction' with potentially interfering nucleic acids or specimen-related conditions) can be determined by interference studies which test 'no template controls' and/or samples containing potentially interfering substances or non-targeted biomarkers. If/when amplification does occur in LAMP assays, various techniques can be used to verify authenticity of the DNA product, and hence analytical specificity of the assay. These include assessment of amplicon specificity using a specific restriction enzyme and agarose gel electrophoresis, amplification and melting curve studies, and nucleic acid sequencing techniques [27, 28]. Assuring analytical specificity is particularly important in assay design because several phenomena including the formation of amplifiable primer-dimers and hairpin structures can lead to false-positivity [29–31].

Diagnostic performance characteristics include diagnostic sensitivity (which is the ability of a test to correctly classify an individual as having a condition or disease, i.e. the number of true positive results as a fraction of the total number individuals with the condition or disease) and diagnostic specificity (which is the ability of a test to correctly classify an individual as not having a condition or disease, i.e. the number of true negative results as a fraction of the total number of individuals without the condition or disease). An important consideration for diagnostic accuracy studies is the choice of reference standard, which is used to compare the performance of the 'index test' against. This may be an alternative 'gold-standard' assay or may be based on validated clinical diagnostic criteria [24].

### Aim

The primary aim of this scoping review was to describe the technical features and performance characteristics of previously developed serotype-specific dengue RT-LAMP assays. The secondary aim was to evaluate their potential for use in portable molecular diagnostic devices.

## Methods

Methods for this scoping literature review were developed according to the Preferred Reporting Items for Systematic reviews and Meta-Analyses extension for Scoping Reviews guideline [32]. A literature search in Medline was conducted on 4th May 2022. The search strategy was constructed by two investigators from combinations of medical subject headings (MeSH) and keywords (Table 1). Results were imported into Covidence software and deduplicated [33]. Citations were reviewed at a title/abstract level for potential inclusion, then at full text level for inclusion by two authors. Discrepancies were resolved by discussion until consensus was reached. Citation lists from all studies examined at full text level, as well as those from all review articles identified by the original search, were also reviewed.

Studies were included if they were listed before 4th May 2022 (no prior time limit set) and fulfilled either of the following criteria: 1) Described one or more newly developed RT-LAMP assay which was designed to detect a single DENV serotype 1–4 ('original assays'); 2) Described the further evaluation, adaption or implementation of one these assays. Studies were excluded if they did not detail the primer sets which were used (either within the publication, supplementary material or by reference), if they described only generic dengue assays, or if they were not written in English language. All types of laboratory or clinical study design were eligible.

Data were collected from included studies according to a pre-determined proforma. This was designed based on the required performance characteristics which are needed before implementation of laboratory-developed tests, as detailed in the Revised Clinical Laboratory Improvement Amendments (CLIA) of 2003 [34] (analytical performance characteristics) and the updated Standards for Reporting Diagnostic Accuracy (STARD) statement of 2015 (diagnostic performance characteristics) [35]. Where LLOD was given in plaque forming units (PFU) or RNA copies per microlitre, this was converted to PFU or RNA copies per reaction (by multiplying by the reported reaction volume). Technical features of the assay (extraction method, reaction mixture ingredients, primer sequences, incubation temperature and detection method) were also collected.

## Results

The database search retrieved 87 unique articles, of which 46 were excluded based on titles and abstracts. Two articles requiring full text review were also identified through review of citations. Forty-three full texts were therefore assessed for eligibility, of which 22 were included

**Table 1. Search terms.**

| |
|---|
| 1 exp Dengue/ or exp Dengue Virus/ |
| 2 dengue.mp. |
| 3 loop mediated.mp. |
| 4 isothermal amplification.mp. |
| 5 LAMP.mp. |
| 6 1 or 2 |
| 7 3 or 4 or 5 |
| 8 6 and 7 |

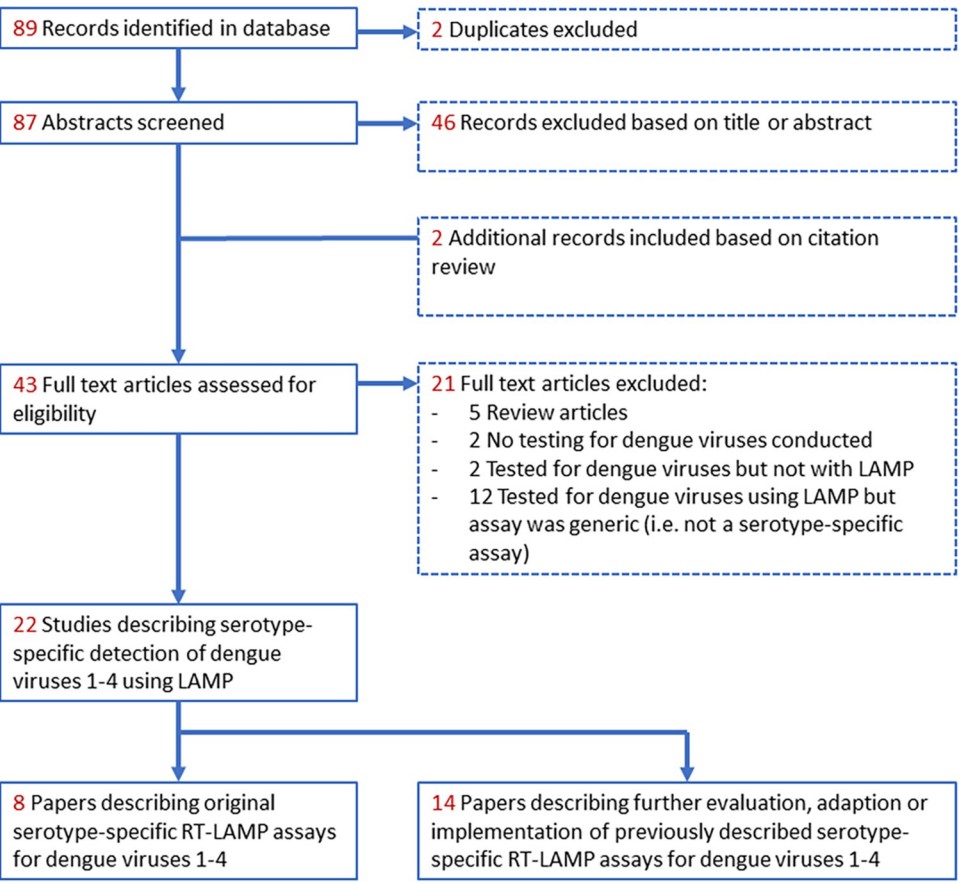

**Fig 1. Workflow showing the assessment of articles and their inclusion in this review.**

and 21 were excluded. The commonest reason for exclusion at full text review was 'assay was generic (i.e. not a serotype-specific assay)', which applied to 12 articles. Of the 22 studies included, 8 described original assays and 14 described the further evaluation, adaption, or implementation of a previously developed assay. A consort diagram is shown in Fig 1.

## Technical features of original RT-LAMP assays

Eight sets of original RT-LAMP assays were developed by Parida et al (2005), Neeraja et al (2015), Hu et al (2015), Lau et al (2015), Yaren et al (2017), Kim et al (2018), Lopez-Jimena et al (2018) and Shoushtari et al (2021) [36–43]. They were designed by obtaining sequences for dengue virus serotypes 1–4 from GenBank/NCBI database (7/8) or other sources (1/8). Various methods and software packages were used to identify potential template regions where sequences were conserved within sequences from the same serotype (but distinct from other serotypes and organisms) and assess possible secondary structures of primers. This included DNASIS software (Hitachi, Japan, 2/8 studies), Primer-Explorer V5 (Eiken Chemical Co. LTD, Tokyo, Japan, 2/8), LAMP designer (Primer biosoft, America, 1/8), Primer-Explorer V3 (Eiken Chemical Co. LTD, Tokyo, Japan, 1/8), OligArch v2 (FfAME, Alachua, FL, 1/8), PrimerCompare v1 (FfAME, Alachua, FL, 1/8), and various R packages (1/8). Fig 2 shows a schematic of the dengue genome and the position of primer-binding for each serotype-specific RT-LAMP assay.

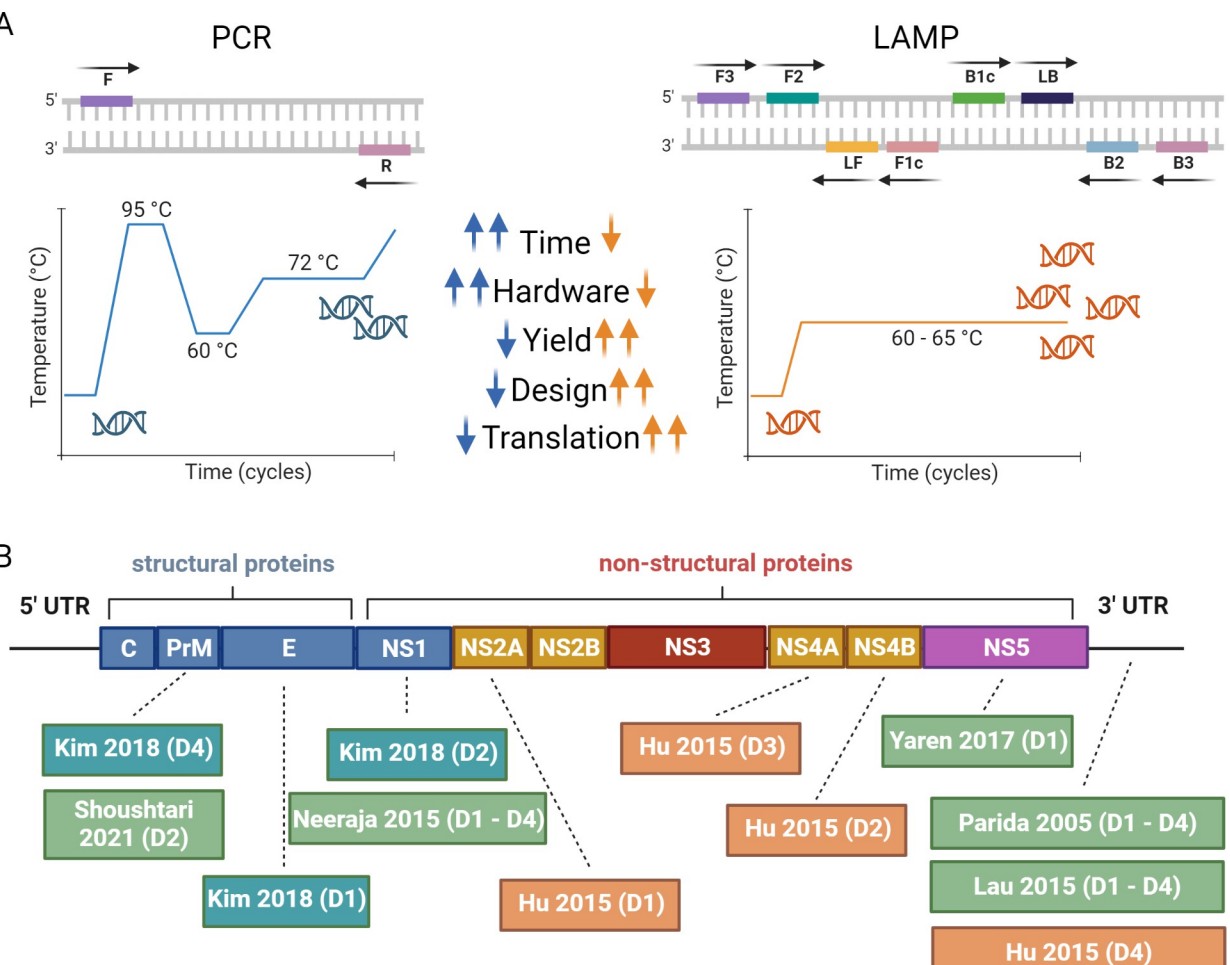

**Fig 2.** (A) Comparison between polymerase chain reaction (PCR) and loop-mediated isothermal amplification (LAMP). Primer binding regions are shown at the top, thermal cycling conditions at the bottom, and differences between the methods in the text where arrows in orange refers to LAMP and in blue to PCR. (B) Schematic of dengue genome showing the position of primer-binding in published serotype-specific RT-LAMP assays.

Out of the eight original assays, seven performed nucleic acid extraction using commercially available kits (most commonly QIAamp viral RNA mini kit, QIAGEN, Hilden Germany). All assays used commercially available preparations of *Bst* DNA polymerase and a reverse transcriptase enzyme (Avian Myeloblastosis Virus, AMV, or RTx reverse transcriptase). Some assays used commercially available LAMP reaction mixes, while others used bespoke mixes which included deoxynucleoside triphosphates, betaine, Tween 20, $(NH4)_2SO4$, $MgSO_4$ (or $MgCl_2$), KCl and Tris–HCl.

## Analytical performance of original RT-LAMP assays

When determined in 7/8 studies, LLOD of assays was between $2.5 \times 10^{-3}$ and $1.22 \times 10^{0}$ plaque-forming unit (PFU)/reaction (for studies which quantified target in PFU) and between $2.2 \times 10^{1}$ and $8.25 \times 10^{2}$ copies/reaction (for studies which quantified target in RNA copies). However, some studies did not give detailed description of the method for quantification of viral particles and/or copies of template RNA used in LLOD experiments. Furthermore, it was sometimes not clear whether the cited concentrations referred to those of original samples, the

elution buffer after nucleic acid extraction (i.e. the extract) or the final reaction mix. In this review, concentrations have been converted to 'per-reaction values', as best possible from the information available in manuscripts. When determined in 3/8 studies, the linear range of quantification went as low as $2.5 \times 10^0$ PFU/reaction (for the study which quantified target in PFU) and as low as $1.98 \times 10^2$ copies/reaction (for studies which quantified target in RNA copies), but similar difficulties interpreting quantification methods and cited concentrations were encountered.

Analytical specificity was usually assessed by testing viral particles of similar viruses (including discordant DENV serotypes and other flaviviruses), or their synthetic RNA or DNA templates. Virtually no incidents of non-specific amplification were reported across all studies. However, the total number of reactions conducted during these interference studies was often not reported, or was fewer than 10. Authenticity of the amplified product was evaluated using digestion with a specific restriction enzyme and agarose gel electrophoresis in 3/8 studies, nucleic acid sequencing in 3/8 studies, melting curve studies in 1/8 study, and was not done in 1/8 study. However, sometimes these data were not presented, and it was rarely clear whether authentication was undertaken for all experiments (i.e. every sample which was determined positive), or only a subset.

## Diagnostic performance of original RT-LAMP assays

Assessment of diagnostic performance occurred in 6/8 studies. Most often panel of 'positive samples' were used, which had been tested in parallel using alternative methods (5/6 studies), or had been characterised previously and assigned 'dengue positive' based on clinical criteria (1/6 studies). A panel of 'negative samples' from healthy individuals were also used in most studies (5/6). RT-qPCR was most often used as a reference standard, but some studies did not clearly detail which test and/or clinical case definition was being used as a reference standard. In 1/6 study RT-LAMP testing resulted in significantly higher positivity than RT-qPCR, which was interpreted as superior diagnostic sensitivity but may also have been due to low diagnostic specificity.

## Adaption of original RT-LAMP assays towards portable molecular diagnostic devices

Fourteen **studies** described the further evaluation, adaption, or implementation of a previously developed assay [29, 44–56]. None of these studies present a working portable molecular diagnostic device which has been deployed and thoroughly evaluated outside the laboratory setting. However, the following technological advances were presented:

Yamagishi et al (2017) adapted the Parida et al assay, analysing samples directly (i.e. without nucleic acid extraction) and loading amplified products into a portable MinION sequencer to determine the serotype. Sequencing was deemed necessary because erroneous LAMP signals were observed from negative control samples, and the workflow was ultimately used under 'field conditions' at a small clinic in Indonesia [47].

Ganguli et al (2017) printed and dried Hu et al primers onto microfluidic channels in bespoke sample-processing and amplification chips. Amplification occurred on the chip while it is housed within a 3d-printed light-proof cradle and a smart phone was used to perform real time detection of fluorescence in each channel [51].

Minero et al (2017) developed and applied two different detection methods using optomagnetic spectroscopy to the Lau et al assays. First, the interaction between biotinylated FIP or LF primers (which incorporate into amplicons during the LAMP reaction) and streptavidin-coated magnetic nanoparticles (included in the reaction mixture) was observed in real-time.

Second, a method was devised to try and discriminate between 'true positive' and 'spurious' LAMP amplicons using a 3'-biotiylated 'loop-validating' DNA probe. In this study the authors highlight the common problem of spurious amplicons in LAMP assays and the importance of having a readout method which is not prone to detection of these [52].

Priye et al (2017) described a 'quenching of unincorporated amplification signal reporters' (QUASR) technique which was used to multiplex the Lau et al DENV1 and DENV2 assays. The BIP primer was labelled with cyanine-5 and a short complimentary quenching probe was included, resulting in fluorescence upon cooling if specific amplification had occurred. The whole workflow was transferred into a 'smart phone-operated LAMP box' which included a heating module, an assay reaction housing module and an optical-detection/image-analysis module and gave a qualitative result for each target [53].

Hin et al (2021) used The Lopez-Jimena assays in an automated device performing sample lysis, nucleic acid extraction, and up to 12 parallel LAMP reactions which are detected in real-time using fluorescence (the 'FeverDisk'). Analytical performance characteristics for the DENV assays in this format were not determined but specimens from some participants were tested positive [56].

Kumar et al (2022) coated Prida et al primers with either biotin or digoxigenin, precipitated the amplified product using polyethylene glycol, and induced clumping with streptavidin- or anti-digoxigenin-coated magnetic particles. This produced a 'magnetic' assay which could be interpreted visually and multiplexed with another assay [48].

Table 2 and S1 Data summarise the studies included in this review.

## Discussion

This review identified eight studies describing original serotype-specific dengue RT-LAMP assays. All assays underwent evaluation of analytical performance with some also undergoing evaluation of diagnostic performance. However, studies were heterogenous in their design and reporting, and some omitted key experimental details. This made objective assessment and comparison of assays difficult and would likely affect attempts to replicate assays and verify findings. It is acknowledged that reports of assay development in academic literature is often a preliminary step, occurring before more rigorous efforts are made to achieve validation and accreditation. As such, authors may not be expected to fulfil requirements such as those set by CLIA for implementation of laboratory-developed tests (which were used as a template for data collection in this study). Nevertheless, standardised assay evaluation and more detailed reporting of performance would be beneficial.

When detailed, the method of nucleic acid extraction, ingredients of the reaction mix (apart from primers) and enzymes were broadly similar across original assays. However, incubation temperature, incubation duration and method for detecting the amplified product varied considerably. LLODs and linear ranges were described for some assays and these analytical performance characteristics were comparable to those which are achievable with many RT-qPCR assays [57]. Analytical specificity was also reportedly good, with virtually no incidents of non-specific amplification being reported. However, the numbers of experiments conducted using no-template controls was generally low, and subsequent studies which used the same primer sets cite non-specificity as a particular reason for modifying incubation settings, primers and/or the detection method. Some studies interpreted higher positivity by RT-LAMP (index test) when compared to RT-qPCR (reference test) as evidence of superior RT-LAMP sensitivity, when in fact this could have been due to lower RT-LAMP specificity. Non-specific reactions are a feature of some nucleic acid amplification assays, including LAMP [29–31]. Therefore, assays must be designed carefully and evaluated thoroughly when this amplification chemistry

**Table 2. Summary of studies describing original RT-LAMP assays for dengue virus serotypes 1–4.**

| Publication (year) | Assay details | | | | Analytical performance | | | | Diagnostic performance | | | | Subsequent publications (year) |
|---|---|---|---|---|---|---|---|---|---|---|---|---|---|
| | Extraction method | Reaction mixture | Incubation | Detection | Target (gene) | Lower limit-of-detection | Specificity: other organisms | Specificity: examination of amplified product | Specimens | Reference standard | Sensitivity | Specificity | |
| Parida et al (2005) | QIAamp viral RNA mini kit (QIAGEN, Hilden Germany) | Loopamp DNA amplification kit (Eiken Chemical Co. Ltd, Japan) | 63.0 degrees Celsius for 60 mins (but determined positive at 30 mins) | Visual inspection (+/- addition of SYBR Green I) and real-time monitoring of turbidity and agarose gel analysis* | DENV1 (3' UTR) / DENV2 (3' UTR) / DENV3 (3' UTR) / DENV4 (3' UTR) | 1 PFU/ml (= 2.5x10^2 PFU/ reaction) / 0.1 PFU/ml (= 2.5x10^3 PFU/ reaction) / 0.1 PFU/ml (= 2.5x10^3 PFU/ reaction) / 0.1 PFU/ml (= 2.5x10^3 PFU/ reaction) | JEV, WNV or SLEV templates—no amplification | Restriction enzyme digestion + agarose gel electrophoresis—product sizes in good agreement with predicted. Further confirmation with sequencing—nucleotide sequences matched target | 83 serum samples: - 25 confirmed dengue cases—38 suspected dengue cases—20 healthy individuals | Considered positive if: Conventional RT-PCR (+) OR nested RT-PCR (+) OR virus isolation (+) | 31/31 (100.0%) | 20/20 (100.0%) | Li et al (2011) Chagan-Yasutan et al (2013) Lo et al (2013) Yamagishi et al (2017) Kumar et al (2022) Gaber et al (2022) |
| Neeraja et al (2015) | QIAamp viral RNA mini kit (QIAGEN, Hilden Germany) | Isothermal Master Mix ISO-001 (Optigene, U. K.) | 63.0 degrees Celsius for 35 minutes | Visual inspection (+/- addition of SYBR Green I) and agarose gel analysis* | DENV1 (NS1) / DENV2 (NS1) / DENV3 (NS1) / DENV4 (NS1) | N/A | Other flaviviruses (sic) including JEV, WNV, HCV and CHIKV—no amplification | Restriction enzyme digestion + agarose gel electrophoresis—product sizes in good agreement with predicted. Further confirmation using sequencing—results not reported in manuscript | 300 serum or plasma samples: - 250 dengue cases- 50 healthy individuals | Considered positive if: RT-qPCR (+) | 140/140 (100.0%)** | 152/160 (95.0%)** | Dave et al (2022) |

*(Continued)*

**Table 2.** (Continued)

| Publication (year) | Assay details | | | | | Analytical performance | | | Diagnostic performance | | | | Subsequent publications (year) |
|---|---|---|---|---|---|---|---|---|---|---|---|---|---|
| | Extraction method | Reaction mixture | Incubation | Detection | Target (gene) | Lower limit-of-detection | Specificity: other organisms | Specificity: examination of amplified product | Specimens | Reference standard | Sensitivity | Specificity | |
| Hu et al (2015) | QIAamp viral RNA mini kit (QIAGEN, Hilden Germany) | Bespoke mix | 63.0 degrees Celsius for 45 minutes | Visual inspection (+/- addition of SYBR Green I) and real-time monitoring of flourescence (SYBR Green I) and agarose gel analysis* | DENV1 (NS2A) | 1x10^1 copies/uL (= 2.5x10^2 copies/ reaction) | JEV, YFV, HSV and Epstein-Barr virus x10 times—no amplification | Restriction enzyme digestion + agarose gel electrophoresis —images shown in manuscript. Further confirmation using sequencing —'specificity of amplification confirmed' | 210 serum samples: - 190 confirmed dengue cases—20 healthy individuals | Considered positive if: 'confirmed to be infected by dengue by clinical diagnosis' | 50/50 (100.0%) | 20/20 (100.0%) | Ganguli et al () |
| | | | | | DENV2 (NS4B) | 1x10^1 copies/uL (= 2.5x10^2 copies/ reaction) | | | | | 59/60 (98.3%) | 20/20 (100.0%) | |
| | | | | | DENV3 (NS4A) | 1x10^1 copies/uL (= 2.5x10^2 copies/ reaction) | | | | | 40/40 (100%) | 20/20 (100%) | |
| | | | | | DENV4 (3'UTR) | 1x10^1 copies/uL (= 2.5x10^2 copies/ reaction) | | | | | 39/40 (97.5%) | 20/20 (100%) | |
| Lau et al (2015) | QIAamp viral RNA mini kit (QIAGEN, Hilden Germany) | Loopamp RNA amplification kit (Eiken Chemical Co. Ltd, Japan) | 65.0 degrees Celsius for 30 min (DENV1-3 assays) or 45 min (DENV4 assay) | Visual inspection (+ HNB dye, Sigma, USA) and real-time monitoring of turbidity* | DENV1 (3' NCR) DENV2 (3' NCR) DENV3 (3' NCR) DENV4 (3' NCR) | The detection limit of RT-LAMP for 3'-NCR was as low as ten copies (= 2.5x10^2 copies/ reaction) | JEV, CHIKV and Sindbis virus—no amplification | Agarose gel electrophoresis —typical DNA ladder observed (though analysis of amplicon size using restriction enzyme not reported) | 213 serum samples— 189 suspected dengue cases—24 healthy individuals | Considered positive if 2 or more of the following were true: RT-qPCR (+), ELISA (+), RT-LAMP (+). | 115/115 (100%) | 98/98 (100%) | Minero et al (2017) Priye et al (2017) Meagher et al (2018) Sigera et al (2019) |
| Yaren et al (2017) | Unclear | Bespoke mix | 65.0 degrees Celsius for 60–90 minutes | Real-time detection of fluorescence (TAMARA-labelled LB or LF probe) then modification to include 'target specific strand-displaceable probe' (fluorescence detected by cell phone camera). | DENV1 (NS5) | 1.22 PFU per assay (= 1.22x10^0 PFU/ reaction) | ZIKV and CHIKV RNA —no amplification | Agarose gel electrophoresis —typical DNA ladder observed (though analysis of amplicon size using restriction enzyme not reported) | N/A | N/A | N/A | N/A | Yaren et al (2018) |

(*Continued*)

**Table 2.** (Continued)

| Publication (year) | Assay details | | | | | Analytical performance | | | | Diagnostic performance | | | | | Subsequent publications (year) |
|---|---|---|---|---|---|---|---|---|---|---|---|---|---|---|---|
| | Extraction method | Reaction mixture | Incubation | Detection | Target (gene) | Lower limit-of-detection | Specificity: other organisms | Specificity: examination of amplified product | Specimens | Reference standard | Sensitivity | Specificity | | | |
| Kim et al (2018) | QIAamp viral RNA mini kit (QIAGEN, Hilden Germany) | Bespoke mix | 69.7 (DENV1 assay), 65.0 (DENV2 assay) or 66.5 (DENV4 assay) degrees Celsius for 40 minutes | Visual inspection (UV light illumination) | DENV1 (E) | 33 copies / uL (= 8.25x10^2 copies/ reaction) | DENV3, norovirus, rotavirus and bovine viral diarrhea—no amplification | Agarose gel electrophoresis —typical DNA ladder observed (though analysis of amplicon size using restriction enzyme not reported) | N/A | N/A | N/A | N/A | | | |
| | | | | | DENV2 (NS1) | 3.55 copies / uL (= 8.88x10^1 copies/ reaction) | | | | | | | | | |
| | | | | | DENV4 (PrM) | 9.06 copies / uL (2.27x10^2 copies/ reaction) | | | | | | | | | |
| Lopez-Jimena et al (2018) | Various commercially available extraction methods depending on source of samples/ viruses | Bespoke mix | 64.0 degrees Celsius for 45 min (DENV1 assay), 90 min (DENV2 assay), 75 min (DENV3 assay) or 50 min (DENV4 assay) | Real-time detection of fluorescence | DENV1 (various) | 22 RNA molecules per reaction (= 2.2x10^1 copies/ reaction) | ZIKV, YFV, WNV, Ntaya virus, S. typhi, S. paratyphi, S. pneumoniae and P. falciparum— no amplification | Melting curve analysis—single peak temperatures indicated specific amplification | 78 samples: - 42 imported blood/ serum samples— 36 imported RNA extracts | Considered positive if RT-qPCR (+) | Initially sensitivity = 17/ 24 (70.8%), then false-negative samples re-extracted re-run and sensitivity = 23/ 24 (95.8%)** | 7/7 (100%) ** | | | Hin et al (2021) |
| | | | | | DENV2 (various) | 542 RNA molecules per reaction (= 5.42x10^2 copies/ reaction) | | | | | | | | | |
| | | | | | DENV3 (various) | 92 RNA molecules per reaction (9.2x10^1 copies/ reaction) | | | | | | | | | |
| | | | | | DENV4 (various) | 197 RNA molecules per reaction (= 1.97x10^2 copies/ reaction) | | | | | | | | | |

(Continued)

**Table 2.** (Continued)

| Publication (year) | Assay details | | | | | Analytical performance | | | | Diagnostic performance | | | | Subsequent publications (year) |
|---|---|---|---|---|---|---|---|---|---|---|---|---|---|---|
| | Extraction method | Reaction mixture | Incubation | Detection | Target (gene) | Lower limit-of-detection | Specificity: other organisms | Specificity: examination of amplified product | Specimens | Reference standard | Sensitivity | Specificity | | |
| Shoushtari et al (2021) | QIAamp viral RNA mini kit (QIAGEN, Hilden Germany) | Bespoke mix | 65.0 degrees Celsius for 60 minutes | Agarose gel analysis | DENV2 (C-PrM) | 100 RNA copies per reaction (= 1x10^2 copies/ reaction) | DENV1, DENV3, DENV4, WNV, YFV, ZIKV RNA (and serum from hepatitis C patient)— no amplification | N/A | 31 serum samples— 20 dengue cases—11 healthy individuals | Considered positive if RT-qPCR (+) | 15/15 (100%) | Results for 11 healthy sera not presented | |

Abbreviations: DENV = dengue virus, PFU = plaque forming units, JEV = Japanese encephalitis virus, WNV = West Nile virus, SLE = St Louis encephalitis virus, HCV = hepatitis C virus, CHIKV = chikungunya virus, HSV = herpes simplex virus, ZIKV = Zika virus, RT-PCR = reverse-transcriptase polymerase chain reaction, ELISA = enzyme-linked immunosorbent assay, RT-LAMP = reverse transcriptase loop-mediated isothermal amplification

\* Studies frequently described more than one method for detecting amplified products of RT-LAMP. However, it was sometimes not clear how discrepant results were handled in analysis of assay analytical and diagnostic performance

\*\* Multiple alternative analyses are reported in the manuscript

is used. Further in-silico and in-vitro evaluation and modification of primer-sets may be useful, to inform and ensure their optimal performance in portable molecular diagnostic devices. Any future diagnostic accuracy studies which evaluate RT-LAMP assays (index tests) should clearly state which gold-standard assay or clinical diagnostic criteria (or composite thereof) is being used as a comparator (reference test). RT-qPCR, which is generally considered the highest performing single test for dengue infection, was the most common comparator assay in included studies. However, if novel RT-LAMP based assays are developed which are truly portable and can be used at the 'point-of-care' (including sample preparation, amplification, and detection steps), then diagnostic performance of the system as a whole could also be compared to lateral flow assays, which can be used in similar settings.

Assays went on to be used in 14 subsequent studies. These included studies where samples were tested directly (i.e. without any nucleic acid extraction prior to amplification). Assays which do not need sample preparation would be of huge benefit when considering their translation into portable diagnostic devices. However, the performance of 'direct LAMP' and superiority of LAMP assays over PCR assays in this regard is contentious [58]. They also included studies where reagents were lyophilised or 'printed' into microfluidic channels and those which used novel detection methods including the use of smart phone cameras, electrochemical sensing, and sequencing (MinION). The restriction of this study to include only serotype-specific assays for detecting DENV 1–4 is a limitation, and it is acknowledged that other relevant technological advances are likely to have been made and demonstrated in other applications of LAMP-based diagnostics. Additionally, there may be other data on dengue RT-LAMP assays which have not been published or included in the Medline database or may have otherwise been missed by this scoping review's search strategy.

Overall, findings from this study show that serotype-specific RT-LAMP assays for dengue are high-performing. When coupled with novel methods for sample preparation and detection, these assays may ultimately lead to portable molecular diagnostic devices which could be used across tropical and sub-tropical regions where dengue is endemic.

## Supporting information

**S1 Checklist. PRISMA-ScR checklist.**
(DOCX)

**S1 Data. Data collected from individual papers describing original RT-LAMP assays.**
(XLSX)

## Acknowledgments

This work was supported by the Department of Health and Social Care-funded Centre for Antimicrobial Optimisation (CAMO) at Imperial College London; and the Wellcome Trust. AH and JRM are affiliated with the NIHR Health Protection Research Unit (HPRU) in Healthcare Associated Infections and Antimicrobial Resistance at Imperial College London in partnership with the UK Health Security Agency, in collaboration with, Imperial Healthcare Partners, the University of Cambridge and the University of Warwick. The views expressed in this publication are those of the authors and not necessarily those of the NHS, the National Institute for Health Research, the Department of Health and Social Care, or the UK Health Security Agency. AH is a National Institute for Health Research (NIHR) Senior Investigator and she is Chair of the David Price Evans Global Health and Infectious Diseases Research Group at the University of Liverpool.

This work was also supported by the Research Excellence Development (RED) program, Faculty of Medicine Siriraj Hospital, Mahidol University.

## Author Contributions

**Conceptualization:** Paul Arkell, Dumrong Mairiang, Adisak Songjaeng, Panisadee Avirutnan, Alison Holmes, Jesus Rodriguez-Manzano.

**Data curation:** Paul Arkell, Dumrong Mairiang.

**Formal analysis:** Paul Arkell, Dumrong Mairiang.

**Funding acquisition:** Kerri Hill-Cawthorne, Panisadee Avirutnan, Pantelis Georgiou, Alison Holmes, Jesus Rodriguez-Manzano.

**Investigation:** Paul Arkell, Dumrong Mairiang, Adisak Songjaeng.

**Methodology:** Paul Arkell, Dumrong Mairiang, Adisak Songjaeng, Kenny Malpartida-Cardenas, Alison Holmes, Jesus Rodriguez-Manzano.

**Project administration:** Kerri Hill-Cawthorne, Alison Holmes.

**Resources:** Paul Arkell, Kerri Hill-Cawthorne, Alison Holmes.

**Software:** Paul Arkell.

**Supervision:** Panisadee Avirutnan, Pantelis Georgiou, Alison Holmes, Jesus Rodriguez-Manzano.

**Validation:** Paul Arkell, Adisak Songjaeng, Panisadee Avirutnan.

**Visualization:** Kenny Malpartida-Cardenas.

**Writing – original draft:** Paul Arkell, Dumrong Mairiang.

**Writing – review & editing:** Paul Arkell, Dumrong Mairiang, Adisak Songjaeng, Kenny Malpartida-Cardenas, Kerri Hill-Cawthorne, Panisadee Avirutnan, Pantelis Georgiou, Alison Holmes, Jesus Rodriguez-Manzano.

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
