## [Decision Letter · Decision Letter 0]

9 May 2023

PGPH-D-22-02107

Analytical and diagnostic performance characteristics of reverse-transcriptase loop-mediated isothermal amplification assays for dengue serotypes 1-4: a scoping review to inform potential use in portable molecular diagnostic devices

Dear Dr. Arkell,

Thank you for submitting your manuscript to PLOS Global Public Health. After careful consideration, we feel that it has merit but does not fully meet PLOS Global Public Health’s publication criteria as it currently stands. Therefore, we invite you to submit a revised version of the manuscript that addresses the points raised during the review process.

We look forward to receiving your revised manuscript.

Kind regards,

Gathsaurie Neelika Malavige

Academic Editor

Journal Requirements:

2. Please provide separate figure files in .tif or .eps format.

Additional Editor Comments (if provided):

Reviewers' comments:

Reviewer's Responses to Questions

**Comments to the Author**

1. Does this manuscript meet PLOS Global Public Health’s publication criteria? Is the manuscript technically sound, and do the data support the conclusions? The manuscript must describe methodologically and ethically rigorous research with conclusions that are appropriately drawn based on the data presented.

Reviewer #1: Yes

Reviewer #2: Yes

2. Has the statistical analysis been performed appropriately and rigorously?

Reviewer #1: Yes

Reviewer #2: N/A

3. Have the authors made all data underlying the findings in their manuscript fully available (please refer to the Data Availability Statement at the start of the manuscript PDF file)?

Reviewer #1: Yes

Reviewer #2: Yes

4. Is the manuscript presented in an intelligible fashion and written in standard English?

Reviewer #1: Yes

Reviewer #2: Yes

5. Review Comments to the Author

Reviewer #1: Greetings!! Method in Abstract: Study design and duration of the study are absent. The authors also should mention the duration of the study in method in the main article. There are some grammatical mistake in some lines otherwise the article is nicely written.

Reviewer #2: 1. This is a very useful paper. But the text is excessive. The manuscript needs to be condensed some more. The word count now is nearly 6000 and this is too much considering the subject matter.

2. It would be helpful for the reader to have a diagram that explains the thrid section - LAMP as a potential solution - and shows how it is different from RT-PCR.

6. PLOS authors have the option to publish the peer review history of their article (what does this mean?). If published, this will include your full peer review and any attached files.

**Do you want your identity to be public for this peer review?** For information about this choice, including consent withdrawal, please see our Privacy Policy.

Reviewer #1: **Yes: **Dr. Afsana Mahjabin,

MBBS, MPH,

Assistant Professor, Department of Community Medicine,

Monno Medical College, Manikganj.

Reviewer #2: No

---

## [Decision Letter · Decision Letter 1]

21 Jun 2023

Analytical and diagnostic performance characteristics of reverse-transcriptase loop-mediated isothermal amplification assays for dengue virus serotypes 1-4: a scoping review to inform potential use in portable molecular diagnostic devices

PGPH-D-22-02107R1

Dear Dr Arkell,

We are pleased to inform you that your manuscript 'Analytical and diagnostic performance characteristics of reverse-transcriptase loop-mediated isothermal amplification assays for dengue virus serotypes 1-4: a scoping review to inform potential use in portable molecular diagnostic devices' has been provisionally accepted for publication in PLOS Global Public Health.

Best regards,

Gathsaurie Neelika Malavige

Academic Editor

Reviewer Comments (if any, and for reference):

Reviewer's Responses to Questions

**Comments to the Author**

1. If the authors have adequately addressed your comments raised in a previous round of review and you feel that this manuscript is now acceptable for publication, you may indicate that here to bypass the “Comments to the Author” section, enter your conflict of interest statement in the “Confidential to Editor” section, and submit your "Accept" recommendation.

Reviewer #1: All comments have been addressed

Reviewer #2: All comments have been addressed

2. Does this manuscript meet PLOS Global Public Health’s publication criteria? Is the manuscript technically sound, and do the data support the conclusions? The manuscript must describe methodologically and ethically rigorous research with conclusions that are appropriately drawn based on the data presented.

Reviewer #1: Yes

Reviewer #2: Yes

3. Has the statistical analysis been performed appropriately and rigorously?

Reviewer #1: Yes

Reviewer #2: N/A

4. Have the authors made all data underlying the findings in their manuscript fully available (please refer to the Data Availability Statement at the start of the manuscript PDF file)?

Reviewer #1: Yes

Reviewer #2: Yes

5. Is the manuscript presented in an intelligible fashion and written in standard English?

Reviewer #1: Yes

Reviewer #2: Yes

6. Review Comments to the Author

Reviewer #1: Greetings! Authors have adequately addressed the comments raised in a previous round of review. So, I think this article is acceptable.

Reviewer #2: I picked out a few small errors and have highlighted them. There may be some I have missed, so please get the manuscript read by fresh eyes.

Why are there no placeholders for tables and figures in the manuscript?

7. PLOS authors have the option to publish the peer review history of their article (what does this mean?). If published, this will include your full peer review and any attached files.

**Do you want your identity to be public for this peer review?** For information about this choice, including consent withdrawal, please see our Privacy Policy.

Reviewer #1: **Yes: **Dr. Afsana Mahjabin

Associate Professor,

Department of Community Medicine,

Monno Medical College, Manikganj.

Reviewer #2: No
